# General-Purpose Coarse-Grained Toughened Thermoset Model for 44DDS/DGEBA/PES

**DOI:** 10.3390/polym12112547

**Published:** 2020-10-30

**Authors:** Michael M. Henry, Stephen Thomas, Mone’t Alberts, Carla E. Estridge, Brittan Farmer, Olivia McNair, Eric Jankowski

**Affiliations:** 1Micron School of Materials Science and Engineering, Boise State University, Boise, ID 83725, USA; mikehenry@boisestate.edu (M.M.H.); stephenthomas1@boisestate.edu (S.T.); monetalberts@u.boisestate.edu (M.A.); 2The Boeing Company, St. Louis, MO 63134, USA; Carla.E.Reynolds@boeing.com (C.E.E.); olivia.mcnair@boeing.com (O.M.); 3The Boeing Company, Huntsville, AL 35808, USA; brittan.a.farmer@boeing.com

**Keywords:** epoxy, coarse-grained, glass transition, molecular dynamics

## Abstract

**Abstract:**

The objective of this work is to predict the morphology and material properties of crosslinking polymers used in aerospace applications. We extend the open-source dybond plugin for HOOMD-Blue to implement a new coarse-grained model of reacting epoxy thermosets and use the 44DDS/DGEBA/PES system as a case study for calibration and validation. We parameterize the coarse-grained model from atomistic solubility data, calibrate reaction dynamics against experiments, and check for size-dependent artifacts. We validate model predictions by comparing glass transition temperatures measurements at arbitrary degree of cure, gel-points, and morphology predictions against experiments. We demonstrate for the first time in molecular simulations the cure-path dependence of toughened thermoset morphologies.

**Dataset:**

https://drive.google.com/drive/folders/12gMdFHTKY72EbIEH1FxxG6dT5cf-i-ac?usp=sharing

**Dataset License:**

CC-BY-SA

## 1. Introduction

Lightweight composites are increasingly used as alternatives to metal components of aircraft, especially over the last decades. Initially reserved for the most demanding aerospace applications, such as fighter aircraft, composite components are now prevalent in commercial aircraft, including 50% of the weight of the Boeing 787 [1]. This proliferation is enabled by improvements in composite formulations and processing, yet there exist significant opportunities to improve the reliable manufacturing of composite aerospace parts. Specifically, control of the thermoset matrix nanostructure (*morphology*) during the curing is currently underdeveloped and improvements could drastically increase the reliability and reduce the time and energy costs of part fabrication [2,3,4]. The challenge lies in understanding how morphology depends on the conditions experienced by the part during curing, and which morphologies have sufficient material properties for specific applications. Improved ability to predict properties from morphologies and morphologies from processing will enable:Predicting how deviations from process specifications impact performance.Composite formulations optimized for manufacturing requirements.Temperature schedules (termed *cure profiles*) optimized for speed and reproducibility.

Embedding fibers in a matrix of polymers serves to support the fibers and transfers loads between them, providing the attractive bulk mechanical properties of fiber-based composites. The main chemical components of a thermoset are an epoxy species, an amine species, and sometimes a toughening agent. Here we focus on the epoxy bisphenol A diglycidyl ether (DGEBA), amine 4,4’-diaminodiphenyl Sulfone (44DDS) mixed with toughener Poly(oxy-1,4-phenylsulfonyl-1,4-phenyl) (PES), a toughened thermoset found in aerospace applications (Figure 1). Thermoset manufacturers recommended cure profiles for matrix formulations based on cure requirements of the crosslinked polymer. Recommended cure profiles are empirically determined and are not necessarily the most efficient paths to sufficiently cured parts.

During curing, the crosslinking of DGEBA and 44DDS lowers the miscibility of PES, and this reaction-induced phase separation (RIPS) results in toughener-rich domain formation [5,6,7,8,9]. Early work by Sultan et al. (Reference [10]) used rubber additives to improve fracture toughness in exchange for lower thermal stability at high temperatures [6]. Since then, control over the toughener domains has been shown to increase fracture toughness without sacrificing other desirable mechanical properties [7,8,11,12,13,14,15,16,17]. The toughener domains improve mechanical properties though a variety of mechanisms, including crack tip blunting, voiding at the interface between thermoset and toughener, and shear yielding [18,19]. Smaller domain sizes are argued to improve mechanical properties, as it results in higher surface area between the thermoplastic and thermoset domains [7]. Block copolymers have also been deployed to control toughener morphology and composite mechanical properties [20,21,22]. Regardless of mechanism, understanding and controlling the morphology of tougheners whose phase-separation is induced by the crosslinking is central to controlling the mechanical properties of the matrix.

Temperature deviations away from a desired cure profile increase the probability that the morphology and material properties of a part are compromised, and these parts must undergo material review to confirm whether this is the case. Material review involves the creation of a sample volume cured with the same temperature deviation as the original part, which then undergoes mechanical testing. Throwing away the deviant part and curing a new one usually costs less time and effort than replicating the deviation and validating the sample volume, which is wasteful in the cases of sufficiently strong deviants. Avoiding this waste would be possible if the sensitivity of mechanical properties to cure profile deviations were more fully understood.

Computer simulations are needed for making sense of cure profile sensitivity because the parameter space combinatorics prohibit experimental enumeration, compounded by the impracticality of obtaining atomic-level detail of each cured morphology. Formulating a thermoset includes choosing the chemistry and proportions of epoxy, crosslinker, toughener, and additives compounds, resulting in combinatorial explosion of candidate formulations. Further, each formulation can result in a wide range of morphologies that depend upon cure profile, the number of which adds another factor to the intractability of enumeration. Models for thermoset curing implemented in computer simulations provide a proxy for part fabrication that are faster and less expensive to perform, and can provide insight into how atomic-level structure evolves and impacts properties. Further, modern GPU (graphics processing unit) hardware enables sensitivity analysis and optimizing cure profiles for desired morphologies because screenings of independent formulations and cure profiles can be performed in parallel.

Computationally predicting morphology requires models that faithfully capture the thermodynamics and kinetics of the crosslinking reaction between amine and epoxy molecules, and resulting phase separation of any tougheners present. Doing so is challenging because reactions dynamics occur at fast ( $1\times 10^{-12}$
s) and small ( $1\times 10^{-10}$
m) scales, while morphology evolution occurs at slow ( $1\times 10^{2}$
s) and large ( $1\times 10^{-6}$
m) scales. Accurately simulating the cross-linking of the epoxy and amine species is crucial when modeling these systems as the bonding network influences the properties of the thermoset [23,24], in particular the relationship between the glass transition temperature $T_{g}$ and cure fraction α described by the DiBenedetto equation [23,25,26,27,28,29,30,31]. Atomistic molecular dynamics (MD) simulations with temperature-independent bonding models have been successfully deployed to generate crosslinked nanostructures and glass transition temperatures $T_{g}$, but are limited to simulation volumes around (13 ${\mathrm{nm}}^{3}$) [32,33,34,35,36]. The work of Li, Strachan and coworkers [32,33] demonstrates atomistic simulations of DGEBA reacted with 44DDS, 33DDS, and other crosslinkers to predict mechanical properties including $T_{g}$, density, modulus, and expansion coefficients. In the case of $T_{g}$ for 44DDS/DGEBA, the atomistic simulations performed overpredict $T_{g,sim}=525$ K compared to DSC experiments $T_{g,exp}=450$ K at 92% cure, though no empirical fitting is performed and cooling-rate-dependent corrections help explain the discrepancy [32,33]. Khare and Phelan investigate similar, untoughened DGEBA (2-mers) and 44DDS and predict 489 K$\leq T_{g,sim}\left(\alpha =100\%\right)\leq 556$ K, depending on cooling rate [36].

Coarse-grained (CG) approaches demonstrate the ability to access substantially larger simulation volumes and time scales than atomistic approaches, and mapping atomistic degrees of freedom into crosslinked networks enables calculation of material properties [37,38,39,40]. In both References numbers [38] and numbers [40], one-site dissipative particle dynamics (DPD) models are used to represent reacting monomers of 44DDS/DGEBA and DGEBA/DETA (Diethylenetriamine), respectively. In both cases, experimentally reasonable $T_{g}$ are calculated after backmapping, and the case is made for large system sizes for observing toughener microstructure [38] and sufficient structural relaxation [40]. Langeloth et al. develop a coarse-grained model of intermediate resolution to study toughened DGEBA/DETA and show significant discrepancies in $T_{g}{\left(\alpha \right)}_{CG}<T_{g}{\left(\alpha \right)}_{AA}$. Earlier this year Pervaje et al. develop another intermediate-resolution coarse-grained model of reacting thermosets parameterized by SAFT-γ Mie calculations, which includes temperature-dependent reactions and a novel bonding algorithm [41]. Applied to polyester-polyol resins, $T_{g}$ predictions from the coarse model are in agreement with experiments [41]. While the exact details and experimental validations depend on the themoset formulation and the force fields used, multiscale approaches that use coarse models to access long times, large volumes, and high cure fractions 0.9<α<0.95 and atomistic simulations for mechanical property calculations have begun spanning the ≈12 orders of magnitude between reaction dynamics and phase separation.

However, to predict how thermoset microstructure depends on cure profiles, temperature-dependent reaction models are necessary. In our prior work developing *epoxpy* [42], we implemented such a reaction model with DPD coarse-grained simulations. Here, we extend *epoxpy* and focus on simulation workflows for parameterizing, validating, and exploring materials behaviors of reacting thermosets with 44DDS/DGEBA toughened with PES as a case study. While prior studies [32,33,36,38,39,40,41,43,44] have included or implemented (1) Reaction rates calibrated against experimentally observed reaction models, (2) Microphase separation of toughener, or (3) $T_{g}\left(\alpha \right)$ validated against experiments, this work is distinguished by the inclusion of all three simultaneously, and crucially (4) We demonstrate for the first time structural sensitivity to cure profile.

## 2. Model

Spherical simulation elements (“beads”) are used to represent monomers of amine 44DDS (A), epoxy DGEBA (B), and each repeat unit of PES (C) 10-mers (Figure 1). Non-bonded interactions are modeled with the 12-6 Lennard-Jones (LJ) potential
$$\begin{array}{ccc} V_{\mathrm{LJ}}\left(r\right)= & 4\varepsilon \left[{\left(\frac{\sigma }{r}\right)}^{12}-{\left(\frac{\sigma }{r}\right)}^{6}\right] & r<r_{\mathrm{cut}}\\ = & 0 & r\geq r_{\mathrm{cut}}, \end{array}$$
where the parameters σ represent “size” of simulation elements and ε sets the magnitude of the potential energy minimum between two simulation elements. Throughout this work σ is used as the dimensionless length scale and $\sigma_{A}=\sigma_{B}=\sigma_{C}=\sigma =1$ nm. We note that the relatively hard-core repulsion of the LJ potential prevents chain crossing that is commonplace in DPD simulations, with impacts on network structure and $T_{g}$ calculations. Energy scales ε calculated from cohesive energy calculations described in Section 4.1.1 and are summarized in Table 1. Interactions between dissimilar simulation elements (“cross” interactions) are obtained using Lorentz-Berthelot (LB) mixing rules applied in prior DGEBA studies [45,46,47], where
(1)$$\epsilon_{AB}=\sqrt{\epsilon_{A}\epsilon_{B}}$$
and
(2)$$\sigma_{AB}=\frac{\sigma_{A}+\sigma_{B}}{2}.$$

Harmonic potentials are used to model bond stretching between pairs of bonded simulations elements. Harmonic angle potentials are used to model bending among triplets of bonded PES (type C) simulation elements, but no angle potentials are used for epoxy-amine triplets. No dihedral or improper constraints are implemented here.

Bond formation between amine and epoxy simulation elements is modeled through the stochastic creation of harmonic bonds between A and B beads that are sufficiently close by an activated process with probability of bond formation
(3)$$p=e^{-\frac{E_{a}Y}{k_{B}T}},$$
where $E_{a}$ is activation energy and bond-order factor Y=1.0 if the bond being proposed is the first bond to form for either bead and Y=1.2 otherwise.

By design, the energy scale for modeling pairwise interactions is distinct from the energy scale for modeling bond formation, which are both distinct from the energy scale for modeling vitrification. This modeling choice facilitates the empirical bridging of timescales that is the focus of the present work through exploitation of temperature-time superposition [24]. We report dimensionless simulation temperatures $T=\frac{k_{B}T^{K}}{\epsilon }$ throughout this work, where $k_{B}$ is Boltzmann’s constant, $T^{K}$ is temperature in Kelvin, and ϵ is an energy unit for either pairwise interactions, bonding reactions, or vitrification. These energy scales span about three orders of magnitude, with $\epsilon_{pair}=\varepsilon =2.1\times 10^{-22}$ J, $\epsilon_{rxn}=1.78\times 10^{-19}$ J, and $\epsilon_{vit}=6.63\times 10^{-21}$ J. The pairwise energy scale is derived from cohesive energy described in Section 4.1.1, the reaction energy scale is set from experimental measurements of activation energy [48], and the vitrification energy scale is set by equating the dimensionless $T_{g}^{sim}\left(\alpha =1\right)$ to an experimental measurement of $T_{g}^{exp}\left(\alpha =1\right)=480$ K [5].

## 3. Methods

Simulations of curing epoxy thermosets (with and without toughener) are implemented with the open source dynamic bonding plugin “dybond” [49] written for the HOOMD-blue [50] molecular dynamics engine. Data storage, retrieval, and job submission is done with the signac [51,52] framework. System initialization is performed with mBuild [53]. Plots are created using matplotlib [54] and all scripts used for job submission and data analysis are available at this repository [55]. We use the bonding algorithm as outlined in our previous work [42]. Briefly, every $\tau_{B}$ molecular dynamics steps we attempt to form $n_{B}$ possible bonds where center-to-center distance between an epoxy and amine simulation element is r≤1.0σ and with probability as in Equation (Equation 4). Here, $n_{B}=0.005n_{T}$, where $n_{T}$ is the total number of bonds that can be formed, equal to four times the number of A beads for the stoichiometric mixtures of A and B. Simulation element positions and velocities are integrated forward in time according to Langevin equations of motion with drag coefficient γ=4.5 and step size δt=0.01. Random initial configurations are used for each independent simulation run. We calculate the toughener (PES-PES, C-C) structure factor S(q) for simulation snapshots using the “diffract” utility described in Reference [56], enabling identification of any periodic domain features that could indicate phase separation. Unless otherwise noted, simulation parameters summarized in Table 2 are used throughout.

Glass transition temperatures are calculated directly from coarse-grained simulation volumes as described in Section 4.3.3 of Reference [57]. Briefly, snapshots of simulations that have reached a specified degree of cure α are used to initialize new simulations that are instantaneously quenched across a range of temperatures to identify $T_{g}$, below which the self-diffusion coefficient D vanishes (Figure 2).

Diffusion coefficients $D=\frac{d\mathrm{MSD}}{6dt}$ are measured directly from quenched trajectories, where MSD is the mean-squared displacement averaged over “B” (DGEBA) simulation elements. We employ piecewise regression to identify the discontinuity in D(T). Calculations of $T_{g}\left(\alpha \right)$ are validated against theory by measuring the R-squared fit of the DiBenedetto equation [58] modifed by Pascault and Williams [31]
(4)$$T_{g}\left(\alpha \right)=\frac{\lambda \alpha (T_{g1}-T_{g0})}{1-\alpha (1-\lambda )}+T_{g0},$$
where λ is chemistry specific and represents the non-linear relationship between $T_{g}$ and degree of cure and varies from 0 to 1 [31], $T_{g0}$ is the glass transition temperature at zero percent cure, and $T_{g1}$ is the glass transition temperature at one hundred percent cure (α=1). We set λ=0.5 for its quality of fit here, and note it is larger than λ from prior work on 44DDS/DGEBA (0.34 [59]–0.38 [60]).

## 4. Results

The 7849 independent MD simulations performed in this work fall into three categories:SetupValidationExploration

In total, approximately 15,000 GPU-hours of simulation time are performed over about three months. Descriptions of analysis and simulation methods specific to each type of simulation are included in the appropriate subsections that follow.

### 4.1. Setup Simulations

We perform 33 all-atom simulations to determine coarse-grained forcefield parameters, 4480 coarse-grained simulations to calibrate reaction kinetics, and 1448 coarse-grained simulations check for finite size effects before peforming validation and exploration studies.

#### 4.1.1. Forcefield Parameterization

We perform 33 all-atom MD simulations to calculate cohesive energies $e_{coh}$ of amine 44DDS (A), epoxide DGEBA (B), and toughener PES (C) moieties to parameterize their non-bonded interactions of their coarse-grained simulation elements $\varepsilon_{i}$. In liquids, $e_{coh}$ represents the energy required to separate molecules from the liquid state into isolated molecules in the vapor phase
(5)$$e_{coh}=E_{bulk}-E_{isolated}$$
and is calculated from the difference in average molar potential energies *E* between bulk and isolated molecules [42,61]. Cohesive energies have been used to estimate macroscopic miscibility [62] and parameterize coarse LJ models [61] and we do the same in the present work. We use the OPLS-2005 force field and NPT simulations at P=1 atm, and simulate 11 temperatures equally spaced over T∈[273,600] K. Each simulation volume is initialized with 500 molecules (monomers of DGEBA and 44DDS, 10-mers of PES) at a density of 1 g/cm${}^{3}$. After equilibration, densities in agreement with experiments of 0.8–1.14 g/cm${}^{3}$ (DGEBA), 1.3–1.1 g/cm${}^{3}$ (44DDS), and 1.3–1.2 g/cm${}^{3}$ (PES) are observed. Averaging over temperatures, we calculate $e_{coh}$ for DGEBA, 44DDS and PES monomers as 30.36 kcal/mol, 27.98 kcal/mol and 26.84 kcal/mol respectively. We de-demensionalizes pairwise interactions in the coarse-grained models by normalizing by the DGEBA cohesive energy, resulting in the interaction potentials of Table 1.

#### 4.1.2. Reaction Kinetics Calibration

Two parameters are tuned to calibrate reaction kinetics: The maximum number of bonds attepted per bonding step $n_{B}$ and the number of time steps between bonding steps $\tau_{B}$. Reaction calibration is important for two primary reasons: First, the higher the ratio of $n_{B}/\tau_{B}$, the faster simulations can cure to higher α, which saves time. Therefore, the largest $n_{B}/\tau_{B}$ that replicates experimental reaction dynamics optimizes computational throughput. Second, validating first-order reaction dynamics lays the foundation for exploratory simulations with self-accelerated reactions. We perform 20 independent coarse-grained simulations of 44DDS/DGEBA/PES at each of 224 combinations of $\left(n_{B},\tau_{B},T\right)$ to identify the combinations that best fit a first-order reaction model from experimental data [48]. Each simulation has N= 50,000 (10,000 A, 20,000 B, and 2000 10-mer chains of C) coarse simulation elements and is cured isothermally at T∈{0.2,0.5,1.0,2,3,4,5,6}. Reaction parameters are sampled over the sets $n_{B}\in \{2.5\times 10^{-5},5\times 10^{-5},1\times 10^{-4},1\times 10^{-2}\}\times n_{T}$ (where $n_{T}$ is the total number of bonds that can be formed, 40,000 here) and $\tau_{B}\in \left\{1,2,10,20,40,80,100\right\}$. We find $n_{B}=2.5\times 10^{-5}n_{T}=1.0$ and $\tau_{B}=1.0$ here, and use $n_{B}=2.5\times 10^{-5}n_{T}$ for other system sizes.

#### 4.1.3. Finite Size Effects

Here we investigate the effect of small system sizes on the prediction of glass transition temperatures and morphology.

#### 4.1.4. Glass Transition—Small Systems

We perform curing simulations and $T_{g}\left(\alpha \right)$ calculations of small N=500 volumes and find deviations relative to N=50,000 predictions of $T_{g}\left(\alpha \right)$. For each N=500 and N=50,000, DGEBA/44DD/PES blends are cured isothermally at T=3. Simulation snapshots at intervals α∈{0,0.3,0.5,0.7} are used to initialize new trajectories that are quenched to T={0.05,0.15⋯,2.95,3.0}. Three independent quenches are performed for each of the 60 quench temperatures. $T_{g}$ calculated from the quenches and the DiBenedetto fits are presented in Figure 3.

While the smaller systems are noisier, the qualitative trend in $T_{g}\left(\alpha \right)$ is not without value, as these predictions can be used for estimates bounds of $T_{g}$ that will lower the computaitonal cost of measuring the glass transition in larger systems.

#### 4.1.5. Morphology—Small Systems

We next apply our model to study the domain sizes of PES toughener that evolve over the course of curing. We use the PES-PES structure factor to quantify the domain size of the PES toughener. We expect sufficiently large system sizes to demonstrate PES domain sizes independent of simulation volume, but to find volumes below which microphase separation cannot be resolved. Throughout this work we use *microphase separation* and *macrophase separation* to distinguish characteristic length scales of the tougheners: In the case of microphase separation, we measure charasteristic spacings of toughener (with a local peak in the structure factor S(q) that are smaller than half the smallest periodic simulation axis $L_{min}/2$) whether or not they or ordered or disordered. In the case of macrophase separation, divergence of S(q) for $q<4\pi /L_{min}$ indicates toughener has aggregated into a domain large enough where microphase separation can no longer be resolved.

Three replicates of system sizes with $N\in \{5\times 10^{4},8\times 10^{4},1\times 10^{5},2\times 10^{5},4\times 10^{5},6\times 10^{5},8\times 10^{5},1\times 10^{6}\}$ are cured isothermally to 90% with fiducial parameters shown in Table 2 and simulations were run for $1\times 10^{7}\;\Delta t$. The resulting structure factors S(q) are summarized in Figure 4 and local maxima in S(q) (red dots) indicate PES domains with a characteristic spacing of 26±2 nm emerge in $N\geq 2\times 10^{5}$ systems.

Importantly, cubic simulation volumes below $N=2\times 10^{5}$ are too small to resolve these 26 nm PES features, as the half-box-length (blue stars) for these volumes are smaller than 26 nm (recall conversion factor $l=\frac{2\pi }{q}$ between lengths *l* and wavenumbers *q*). Note that in the too-small volumes, no local maxima (red dots) are observed, and S(q) appears to diverge at low *q*. Therefore, for studies of microphase separation in 44DDS/DGEBA/PES, system sizes of at least $N=2\times 10^{5}$ are necessary. More broadly, microphase separation on length scales larger than half the periodic box length manifest as macrophase separation because local maxima in S(q) cannot be resolved for $q<\frac{\pi }{L}$ for box length *L*.

### 4.2. Validation Simulations

Validation simulations comprise 1785 coarse-grained MD simulations for calculating gel points, glass transition temperatures, and morphology of toughened 44DDS/DGEBA/PES and untoughened 44DDS/DGEBA blends.

#### 4.2.1. Gel-Point Validation

Isothermal curing simulations of the fiducial N=50,000 toughened 44DDS/DGEBA/PES volumes are performed to predict gelation. The gel-point is dependent on the underlying bonding network that forms as the amine and epoxy react, and is therefore a useful metric for validation in addition to $T_{g}$ and S(q). We calculate the gel-point by examining at what degree of cure α the molecular weight of the largest and second largest chain diverge. We use the NetworkX [63] python package to measure the size of molecules as curing proceeds.

We sample 26 independent isothermally cured (T=3), toughened volumes spanning cure fractions from α=0% to α=92.4% and find the gel-point measured by molecular mass at $\alpha_{gel}=60$% (Figure 5, in good agreement with theory and experiments. Flory-Stockmayer theory of gelation [64,65] predicts that gelation of 44DDS/DGEBA (a bifunctional monomer and a tetrafunctional monomer) at $\alpha_{gel}=58$% [66]. Flory-Stockmayer theory is known to underpredict the cure fraction at gelation, as steric hindrance prevents functional groups reacting with equal probability [67]. Experiments of 44DDS/DGEBA curing measure $\alpha_{gel}>50$% [68] and $\alpha_{gel}=60$% [69].

#### 4.2.2. Glass Transition Validation

A total of 1770 coarse-grained MD simulations are performed to validate predicted $T_{g}\left(\alpha \right)$ against experimental data and theoretical fit to the DiBenedetto equation. First, three independent isothermal curing simulations are performed for N=50,000 systems at the fiducial simulation paramaters. Independent snapshots from α=0 to α=0.9 at intervals of dα=0.1 are taken from each curing simulation to initialize independent quenches (Figure 2). These 30 independent snapshots representing the full range of cure fractions are each quenched in independent simulations to each of the 40 dimensionless temperatures from 0.05 to 2.0 at intervals of dT=0.05, plus each of the 15 temperatures from 2.1 to 3.5 in intervals of dT=0.1, plus T∈{3.6,4.0,4.5,5.0}. From these simulations we focus on α∈{0,0.3,0.5,0.7} for determining fits to the DiBenedetto equation, and temperatures $0.1<T_{quench}<2.5$ for identifying glass transition temperatures.

We use piecewise regression to identify $T_{g}$ from diffusivity measurements from each of the aforementioned simulations (Figure 6a), and fit with the DiBenedetto equation (Figure 6b).

We validate against experiments of 44DDS/DGEBA by setting the extrapolated dimensionless value of $T_{g}\left(\alpha =1\right)=1.32$ equal the experimental measurement 480 K and then checking intermediate α=0.4 predictions. Here, our predicted $T_{g}\left(\alpha =0.4\right)=320$ K is 6.7% higher than the experimental interpolation of 300 K for PES-toughened 44DDS/DGEBA [5], and 6.5% higher than the experimental interpolation of 310 K for the untoughened system [70] (Figure 6b). Several other untoughened epoxy systems which have a similar epoxy/amine chemistry also shows a similar trend in the DiBenedetto equation where the $T_{g}\left(\alpha =0.4\right)\approx 300\;K$ [66,70,71]. It is also known from experiments that the uncured 44DDS/DGEBA/PES system is completely miscible and flows at room temperature. Both conditions ($T_{g}\left(\alpha =0\right)<293$ K, and $T_{g}\left(\alpha =0.4\right)\approx 300$ K) are satisfied by the current model.

#### 4.2.3. Morphology Validation

To validate predictions of microphase separated morphology we first perform 3 independent curing simulations at T=3 of the fiducial simulations (Table 2) at each of 5 system sizes $\left(N=\{4\times 10^{5},\;6\times 10^{5},\;8\times 10^{5},\;1\times 10^{6}\}\right)$. These sizes are chosen because $N=4\times 10^{5}$ corresponds to cubic simulation volumes with side length L=74 nm, far larger than needed to measure 26 nm periodic features with Fourier-based S(q) analysis (see Section 4.1.5). As in the simulations for understanding minimum simulation sizes, we measure the structure factor S(q)–specifically the wave number of any local maxima—to quantify microphase separation and when systems reach steady states. A representative time evolution of S(q) is shown in Figure 7A for an $N=1\times 10^{6}$ system, which reaches steady state after $7\times 10^{6}$ steps.

Figure 7B shows a representative $N=1\times 10^{6}$ morphology after achieving steady state. The average PES-PES S(q) measured for fiducial systems with $N\geq 4\times 10^{5}$ has a local maximum at $q_{max}=0.235\pm 0.020$ nm${}^{-1}$, corresponding to feature spacings of 26.6±2.5 nm.

In experiments by Rosetti et al. [7], chemically similar DGEBF/44DDS toughened with PES is observed to undergo increasing reaction-induced phase separation that increases with increasing cure temperature. Nonfunctional PES, most similar to the system studied here, remains mixed at a cure temperature of 363 K, phease separates into 250 nm domains when cured at 403 K, and 400 nm domains when cured at 423 K. The length scales of nonfunctional PES phase separation we predict here are smaller than those reported in Reference [7], but we observe the same qualitative trend of larger domain sizes with higher cure temperatures in the cure-path-dependent simulations forthcoming in Section 4.3.2. Phenoxy-functionalized PES, which can participate in crosslinking, is observed by Rosetti et al. that smaller PES nodular domains phase separate (40 nm at 4033 K and 150 nm at 423 K). Smaller PES-rich domains are observed in experiments with a tri-functional epoxy, 44DDS, and functionalized PES, around 20 nm [6]. To fully resolve phase separation of 250 nm domains, (500 nm)${}^{3}$ simulation volumes are needed, a factor of 5 larger than the largest volumes cured here. In summary, the simulations presented here demonstrate toughener phase separation on length scales smaller than similar-but-not-equivalent experiments, and $N=1\times 10^{6}$ systems corresponding to (100 nm)${}^{3}$ volumes can routinely be cured to α=0.9 in one week.

### 4.3. Exploration Simulations

Exploration simulations are performed to measure the effect of including reaction enthalpy (80 simulations) and the dependence of cure profile on final morphologies (23 simulations).

#### 4.3.1. Enthalpy Experiment

With temperature-dependent reaction rates in the present model, we perform nonisothermal reaction simulations of otherwise fiducial systems to investigate what models of reaction enthalpy are sufficient for modeling self-accelerated first-order reaction kinetics. In the present case we assume the change in energy associated with the crosslinking reaction is instantaneously distributed among all simulation degrees of freedom, corresponding to an increase in temperature where $\Delta H_{rxn}=C_{v}\Delta T_{rxn}$ for heat capacity $C_{v}$ in the NVT ensembles studied here. We perform simulations with per-bond $\Delta T_{rxn}=0.0,1\times 10^{-6},1\times 10^{-5},1\times 10^{-4}$ in addition to the same $n_{B}$ and $\tau_{B}$ ranges described in Section 4.3.1.

Results summarized in Figure 8 validate first-order reaction kinetics are accurately modeled when $\Delta T\leq 1\times 10^{-6}$, and that $\Delta T=1\times 10^{-4}$ is sufficiently large for self-accelerated first-order kinetics to always beat first-order kinetic fits to concentration profiles. Unlike the isothermal simulation cases where ΔT=0 and reaction kinetics become more accurate as *A* is decreased, in the self-accelerated first-order kinetic models there exist optimal A≈1.

In sum, the present model permits straightforward modeling of self-accelerated reactions through the inclusion of a per-bond change in temperature that is validated against kinetic models.

#### 4.3.2. Sensitivity to Cure Profile

The final studies in this work investigatethe dependence on structure of nonisothermal cure profiles meant to be representative of industrial temperature schedules. We first perform 17 simulations of otherwise fiducial $N=5\times 10^{4}$ volumes that step up from T=2.0 to T=3.5 instantaneously at time $t_{1}$ ranging between $1.5\times 10^{4}$ steps and $4\times 10^{6}$ steps. We next perform 3 replicate simulations of $N=4\times 10^{5}$ volumes that each experience two changes in temperature: From $T_{1}=1.0$ up to $T_{2}=2.0$ at $t_{1}=1\times 10^{5}$ steps, followed by a quench down to $T_{3}=1.2$ at either $t_{2}=2\times 10^{6}$ steps or $t_{2}=9.5\times 10^{6}$ steps. Except for the instantaneous temperature changes described above, the simulations performed in this section are all isothermal. We calculate the time of gelation and S(q) to quantify structure.

Results from the temperature steps from T=2 to T=3.5 are summarized in Figure 9, and demonstrate that gelation before 1e6 steps have elapsed is independent of initial time when $t_{1}<2\times 10^{5}$. Inset in Figure 9b are the cure profiles on semilog axes with open squares indicating gelation times, which are summarized in the main plot.

The delay in gelation with longer times at low *T* is expected because the more time spent at higher temperature, the faster curing occurs, and the faster gelation will occur. Bicontinuous microphase separated morphologies are observed for all simulations here, but no measurable differences in periodic length scales are observed. These results demonstrate that modifying the cure profile enables control over how quickly systems gel.

The final 6 simulations of $N=4\times 10^{5}$ volumes are cured isothermally at $T_{1}=1$ for $1\times 10^{5}$ steps before being instantaneously heated to $T_{2}=2$. Three simulations are quenched to $T_{3}=1.2$ before gelation at $t_{2}=2\times 10^{6}$ steps, and held there until a total of $3\times 10^{7}$ steps have elapsed. The other three simulations are quenched to $T_{3}=1.2$ after gelation at $t_{2}=9.5\times 10^{6}$ steps, and held there until a total of $1\times 10^{7}$ steps have elapsed. Note that $T_{g}\left(\alpha =0.87\right)=1.2$, so systems with α<0.87 will be above the glass transition temperature at all points during these cure profiles. Temperature schedules, gel points, and cure profiles for these pre- and post-gelation quenches are summarized in Figure 10.

The temperature set points correspond to $T_{1}=365\;K$, $T_{2}=730\;K$, and $T_{3}=438\;K$. T2 is chosen such that it is much higher than $T_{g}\left(\alpha =1.0\right)=480\;K$, facilitating diffusion especially before gelation. We analyze morphologies with final cure fraction α=0.855 for both pre-gelation (blue data) and post-gelation (orange) quenches, neither of which is ever below its glass transition temperature.

Average S(q) for the pre- and post-gelation cures are shown in Figure 11.

Two features of the S(q) stand out—first, the length-scales of phase separation are smaller for the pre-gelation quench. Second, there is higher variance in the measured S(q) in the pre-gelation quenches.

The observations of increased phase separation in the post-gelation quench are consistent with experiments demonstrating increased phase separation with higher cure temperatures [5,70]. These observations are also consistent with two different mechanistic explanations: (1) Higher temperatures increases curing rates, which increase reaction-induced phase separation, and (2) Quenching pre-gelation keeps the morphology from being kinetically arrested, and so the tougheners can more easily mix and distribute in the unvitrified volume if thermodynamically favorable. These results demonstrate that thermoset volumes with identical cure fractions can have significant cure-path-dependent microstructures.

## 5. Conclusions and Outlook

We demonstrate a coarse-grained model of toughened epoxy thermosets that
Offers straightforward forcefield parameterization.Can capture first-order and self-accelerated first order reaction dynamics.Is validated against experimental gel points, glass transition temperatures, and morphology for 44DDS/DGEBA/PES blends.Does not require backmapping for $T_{g}$ calculation.Can cure million-particle volumes (corresponding to 31-million atoms and (100 nm)${}^{3}$ periodic boxes) to α=0.9 in under one week.Demonstrates for the first time sensitivity of morphology to cure profile.

To summarize, the present work represents progress towards efficient prediction of the morphology and properties of realistic toughened thermosets and provides template workflows for calibrating models to specific formulations and cure profiles. These functionalities offer opportunity to develop a deeper understanding of aerospace-grade thermosets and more reliable manufacturing processes. As an example, datasets generated here lay the foundation to answer questions about how the degree of phase separation contribute to changes of $T_{g}$ and gelation, which should find applicability beyond the single formulation studied here.

The main shortcomings of this work are the degree of validation against experimental $T_{g}$ and morphology. While the low and high cure fractions matched experimental glass transition temperatures for 44DDS/DGEBA, the curvature of our DiBenedetto fit was smaller than observed in experiments. We expect subsequent work in improved $T_{g}$ detection from diffusivity data, calculation of $T_{g}$ from back-mapped morphologies to provide better predictions of $T_{g}$ across the full spectrum of cure fractions. While we recognize experiments characterizing toughener phase separation on the 10 nm–50 nm length scales are challenging, additional work in this area would provide key datasets to validate against. Alternatively, applying the workflows presented here to thermoset formulations with small-scale phase separation characterized would be a information-rich extension of this work. Finally, this work sets the stage for investigations that simultaneously calibrate the energy scales of monomer interactions, reaction kinetics, vitrification to experimental curing profiles that measure the degree to which hour-long curing profiles can accurately be predicted by a few billion steps of a coarse-grained model.

## Figures and Tables

**Figure 1 polymers-12-02547-f001:**
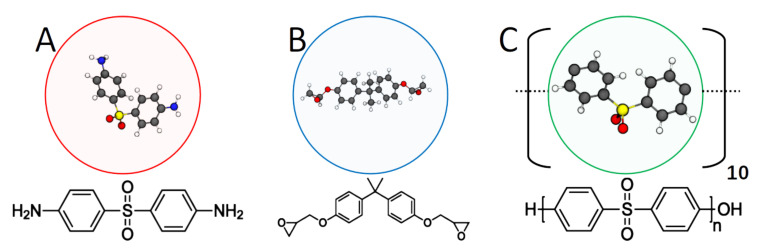
Coarse-grained representations of 44DDS (**A**), DGEBA (**B**), and PES (**C**) repeat units. The amines (**A**) can bond to up to four epoxies (**B**), which can each bond to up to two amines. All toughener molecules are linear 10-mers of (**C**).

**Figure 2 polymers-12-02547-f002:**
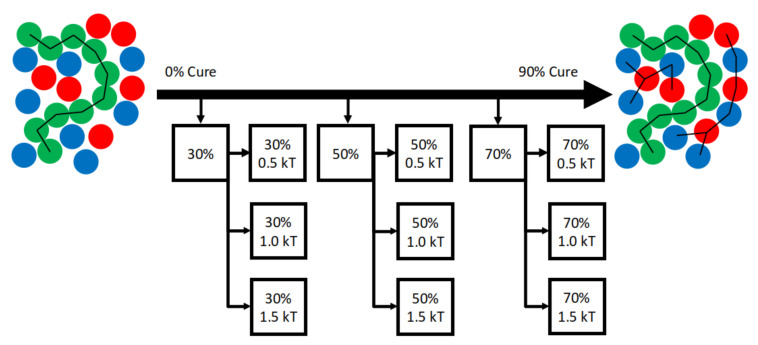
$T_{g}$ prediction workflow: Snapshots at specified α are copied from a curing simulation to initialize instantaneous quenches across candidate low temperatures to identify where the self-diffusion coefficient D vanishes.

**Figure 3 polymers-12-02547-f003:**
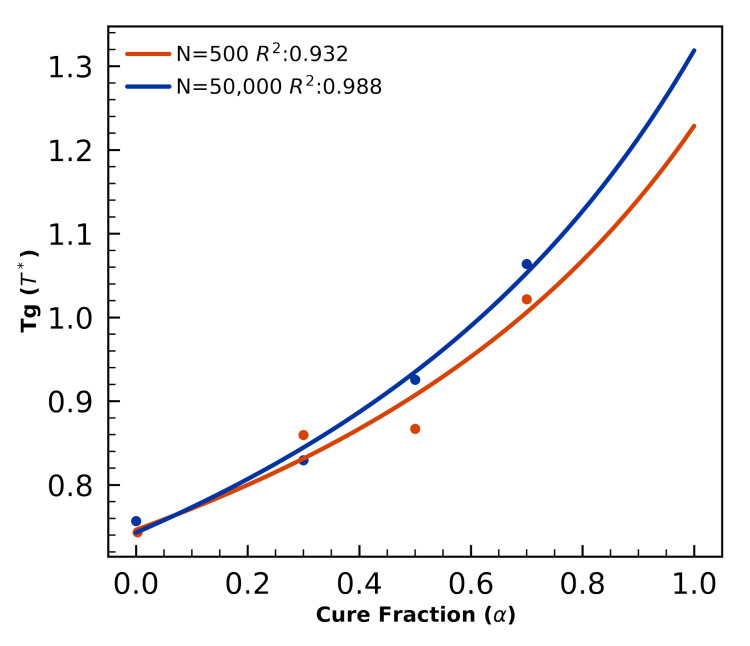
$T_{g}\left(\alpha \right)$ calculations and DiBenedetto fits for N=500 (orange) and N=50,000 volumes of coarse-grained 44DDS/DGEBA/PES show the smaller system sizes result in noiser $T_{g}$ predictions.

**Figure 4 polymers-12-02547-f004:**
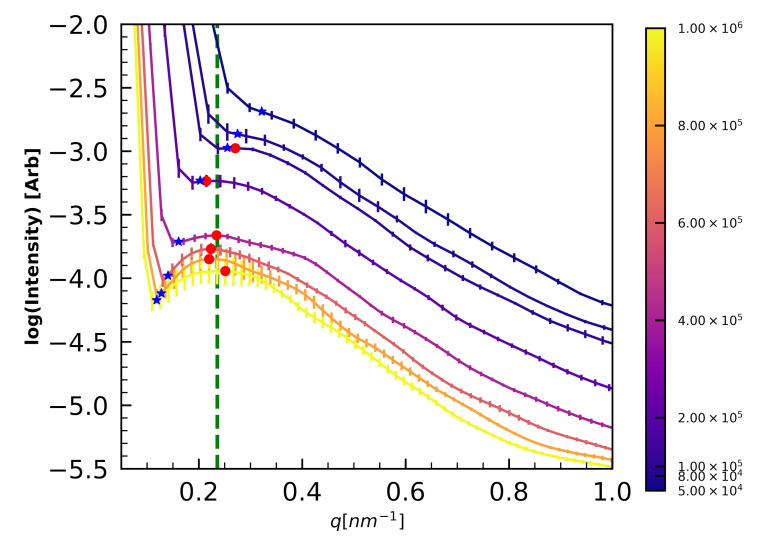
PES-PES structure factor in α=0.9 simulations shows emergence of a 0.236±0.019 nm${}^{-1}$ (26±2 nm) feature (dashed green line), too large to resolve in simulations where $N\leq 2\times 10^{5}$. The color bar indicate system size (*N*). The blue star indicate half of the box length.

**Figure 5 polymers-12-02547-f005:**
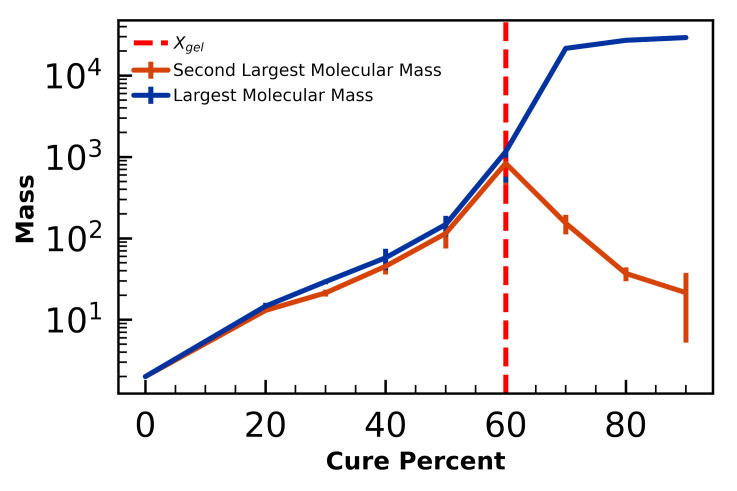
Divergence of the largest (blue) and second-largest (orange) molecular mass indicates gelation, here calculated at α=60%, in agreement with theory (58%) and experiments (60%). Error bars denote standard deviations of 3 independent samples, except the 90% cure case, which have 2 samples.

**Figure 6 polymers-12-02547-f006:**
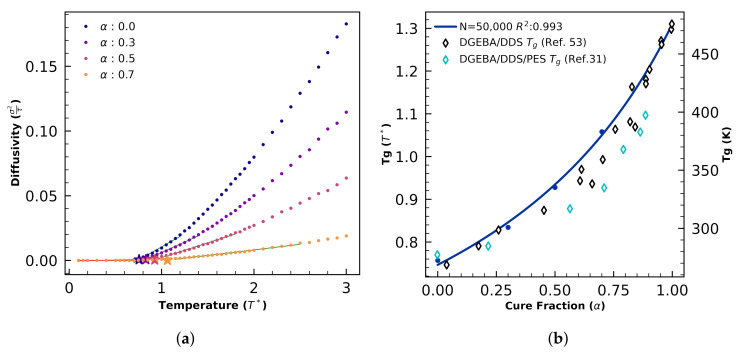
(**a**) Diffusivities measured from quenches of 44DDS/DGEBA/PES as a function of cure fraction and temperature. Green lines indicate linear fits of mid-T diffusivities used to calculate $T_{g}$, which are indicated by stars. (**b**) $T_{g}\left(\alpha \right)$ (blue symbols) and the DiBenedetto fit (blue curve) from (**a**). The simulated $T_{g}$ at low and high cure fractions shows close agreement with $T_{g}$ values measured from an experimental 44DDS/DGEBA system [70] (open black diamonds) and 44DDS/DGEBA/PES [5] (open cyan diamonds).

**Figure 7 polymers-12-02547-f007:**
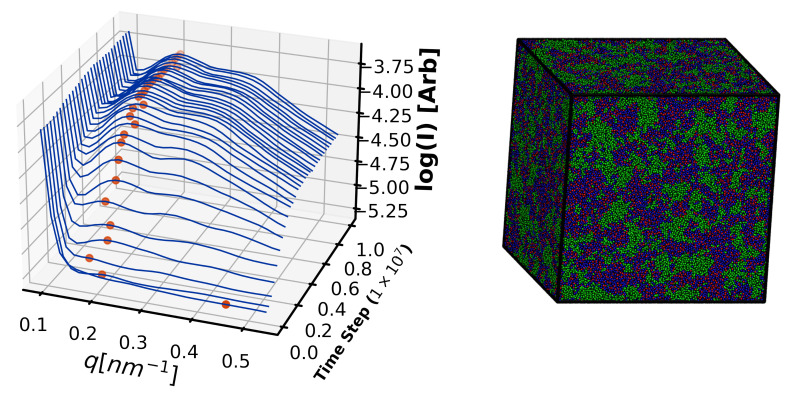
(**A**) Structure factor evolution of PES correlations for $N=1\times 10^{6}$ is used to quantify equilibration. Red symbols indicate the wavenumber $q_{max}$ of a local maximum in S(q). (**B**) representative $N=1\times 10^{6}$ morphology after achieving steady state.

**Figure 8 polymers-12-02547-f008:**
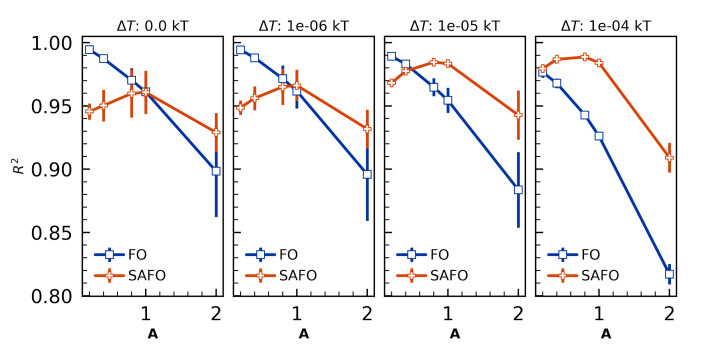
Quality of fit for first-order (FO) and self-accelerated first-order (SAFO) reaction models as a function of $\Delta T_{rxn}$ and $A=\frac{n_{B}}{\tau_{B}}$ validate FO kinetics are most accurate for ΔT=0, and that SAFO kinetics best fit the concentration profiles when ΔT=1e−4. Error bars show standard error in $R^{2}$ value averaged across cure temperatures T=0.5,1.0,2.0,4.0,6.0kT.

**Figure 9 polymers-12-02547-f009:**
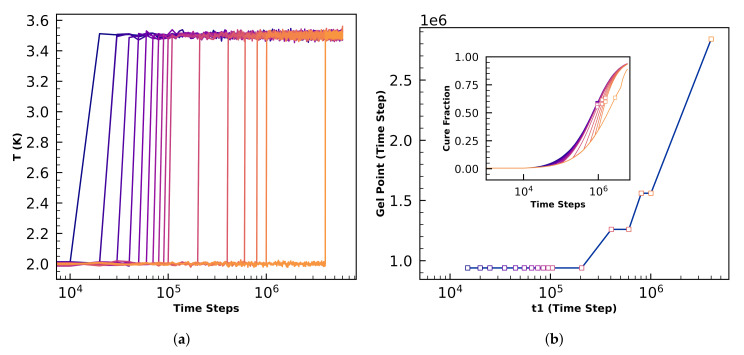
(**a**) Temperature profiles where the initial ramp up time (*t*${}_{1}$) is varied. (**b**) Time to gelation is not affected by *t*${}_{1}<2\times 10^{5}\;\Delta t$. *t*${}_{1}$ time denotes the time at which the cure temperature is ramped up and held constant. Inset in (**b**) are the cure profiles on semilog axes with open squares indicating gelation times.

**Figure 10 polymers-12-02547-f010:**
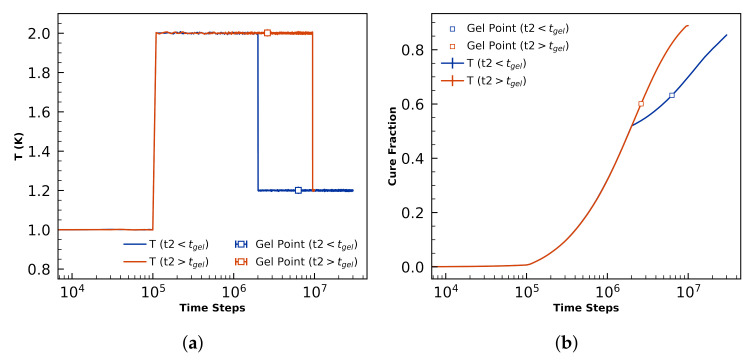
Temperatures profiles (**a**) and curing profiles (**b**) for $t_{2}<t_{gel}$ ($t_{2}=2\times 10^{6}\;\Delta t$) and $t_{2}>t_{gel}$ ($t_{2}=9.5\times 10^{6}\;\Delta t$). The hollow squares show gel point. T2 is chosen to be higher than and T3 is chosen to be slightly lower than the $T_{g}$ of the fully cured system ($T_{g}\left(\alpha =1.0\right)=480\;K$).

**Figure 11 polymers-12-02547-f011:**
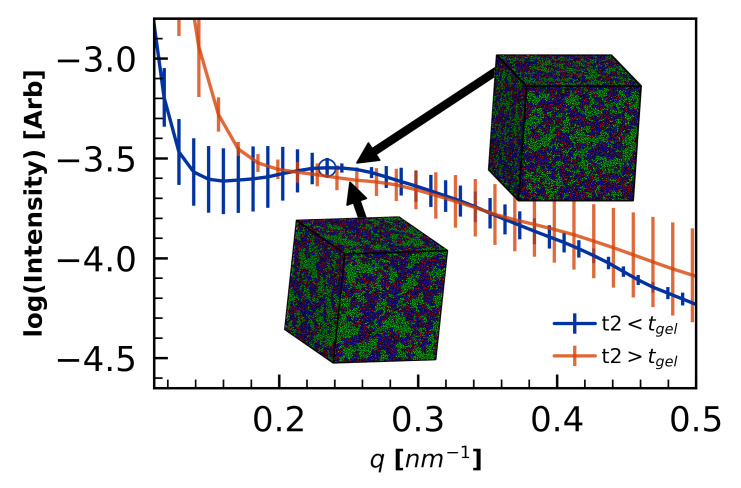
PES-PES structure factor shows difference in morphology as a result of varying *t*${}_{2}$ of the “Step” curing profile. Both simulation volumes are cured to α=0.855. Error bars represent standard error from the three replicate simulations. The length scales of microphase separation are much smaller in the pre-gelation quench (blue), whereas S(q) diverges around $q_{L/2}=0.17$ nm${}^{-1}$, indicating a higher degree of phase separation that is apparent in the more distinct clumping of the inset visualizations.

**Table 1 polymers-12-02547-t001:** Interaction strengths ($\varepsilon_{ij}$) determined by cohesive energy calculations.

	(A) 44DDS	(B) DGEBA	(C) PES
(A) 44DDS	0.9216	0.9600	0.9026
(B) DGEBA		1.0000	0.9402
(C) PES			0.8840

**Table 2 polymers-12-02547-t002:** Fiducial simulation parameters. Note that in the present CG model, monomer% and volume% are equivalent but are not identical to corresponding experimental fractions.

Parameter	Value
Bond equilibrium (A-B,C-C) ($r_{o}$)	1.0 σ
Bond force constant (A-B,C-C) (*k*)	100 $\frac{\epsilon_{pair}}{\sigma^{2}}$
Angle equilibrium (C-C-C) ($\theta_{0}$)	109.5${}^{\circ }$
Angle force constant (C-C-C) ($k_{angle}$)	25 $\frac{\epsilon_{pair}}{\sigma^{2}}$
Non-bonded interaction cutoff $r_{\mathrm{cut}}$	2.5 σ
Number density ($\rho_{n}=N/V$)	1.0
Activation Energy ($E_{A}$)	3.0 $\epsilon_{rxn}$
Bonding distance maximum	1.0 σ
Secondary bond weight (Y)	1.2
Enthalpy of Reaction ($\Delta T_{rxn}$)	0.0
Bond Period ($\tau_{B}$)	1.0
Maximum attempted bonds ($n_{b}$)	$0.005\;n_{T}$
Langevin drag (γ)	4.5
%monomers 44DDS:DGEBA:PES	20:40:40
Cure temperature (*T*)	3.0
Step size (δt)	0.01

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
