# Peer review of "General-Purpose Coarse-Grained Toughened Thermoset Model for 44DDS/DGEBA/PES"

_polymers, 2020, doi:10.3390/polym12112547_

Round 1
Reviewer 1 Report
The manuscript entitled, “General-purpose coarse-grained toughened thermoset model for 44DDS/DGEBA/PES” by Henry et al. involves development of a model that can predict the morphology and the properties of crosslinking polymeric materials used in aerospace applications. They used molecular simulations to find out the cure path dependence of toughened thermoset morphologies. This model will be very useful to control the morphology of the thermoset during the curing process improving the reliability and the part fabrication.
Reviewer 2 Report
In this manuscript, the authors reported a prediction of glass transition temperature, gel point and morphologies of the thermosetting blends composed of bisphenol A epoxy, 4,4-DDS and poly(ether sulfone). By extending the open-source dybond plugin for HOOMD-Blue to implement a new coarse-grained model, the authors determined the parameters for the coarse-grained model from atomistic solubility data, calibrated reaction dynamics against experiments and then checked for size-dependent artifacts. This is an interesting and important work, which should be published in this journal after the authors considered the following minor points:
- In the section of Introduction, the authors should increase the sentences to review the main results of the studies on this system. To my knowledge, there have been a great number of reports on this system involving with reaction-induced phase separation, fracture toughness improvement. A compressive summary is required;
- The term “microphase separation” should be used as “phase separation” because in literature “microphase separation” has a different meaning.
- Please add some discussion about the effect of reaction-induced phase separation on the computation of Tg and gel point in your prediction.
